# TRPA1-Related Diseases and Applications of Nanotherapy

**DOI:** 10.3390/ijms25179234

**Published:** 2024-08-26

**Authors:** Dongki Yang

**Affiliations:** Department of Physiology, College of Medicine, Gachon University, Incheon 21999, Republic of Korea; dkyang@gachon.ac.kr

**Keywords:** TRP channels, TRPA1, nano-delivery systems, nanomedicines

## Abstract

Transient receptor potential (TRP) channels, first identified in *Drosophila* in 1969, are multifunctional ion channels expressed in various cell types. Structurally, TRP channels consist of six membrane segments and are classified into seven subfamilies. Transient receptor potential ankyrin 1 (TRPA1), the first member of the TRPA family, is a calcium ion affinity non-selective cation channel involved in sensory transduction and responds to odors, tastes, and chemicals. It also regulates temperature and responses to stimuli. Recent studies have linked TRPA1 to several disorders, including chronic pain, inflammatory diseases, allergies, and respiratory problems, owing to its activation by environmental toxins. Mutations in TRPA1 can affect the sensory nerves and microvasculature, potentially causing nerve pain and vascular problems. Understanding the function of TRPA1 is important for the development of treatments for these diseases. Recent developments in nanomedicines that target various ion channels, including TRPA1, have had a significant impact on disease treatment, providing innovative alternatives to traditional disease treatments by overcoming various adverse effects.

## 1. Introduction

Transient receptor potential (TRP) channels, identified through studies of *Drosophila* mutants with defective light sensing, are a large group of ion channels expressed on the membranes of various cell types and perform multiple functions [1]. Although TRP channels have been the subject of interest for many researchers for a long time, their significance remains unclear. Approximately 28 types of TRP channels with similar characteristics have been identified in various organisms [2]. A detailed analysis of the structural similarity revealed that the protein encoded by *trp* is a non-selective cation channel with six diaphragm segments [3]. TRP channels are organized into seven subfamilies on the basis of their amino acid sequences [4]. The TRPC family (where “C” is “canonical”) is a very close homolog of the *Drosophila* TRP channel. The TRPC family, which includes channels TRPC1 to TRPC7, has high calcium ion permeability among the non-selective cation channels, and all isotypes, except TRPC2 (a pseudogene), are expressed in humans [5]. The TRPV family (where “V” is “vanilloid”) has vanilloid receptor 1 (TRPV1). The TRPV family consists of multimode ion channels activated in response to heat, acid, mechanical stress, osmotic pressure, and thermal stimulation [6]. The TRPM family (where “M” is “melastatin”) performs a variety of functions in vivo, including redox, insulin secretion, inflammation, and temperature sensing, and it is activated by lipids, ions, and small molecules [7]. The TRPML family (where “ML” is “mucolipin”) has been implicated as an essential element in endolysosomal function and is involved in pH regulation, lysosomal exocytosis, endolysosome fusion/fission and transport, and autophagy [8]. The TRPP family (where “PP” is “polycystin”) is transformed into a receptor-channel complex by polycystin-1 family proteins, and it performs a major function by being involved in a signal transduction pathway that includes calcium as a secondary messenger [9]. The TRPA family (where A is “ankyrin”) contains an ankyrin repeat domain not found in other families. Finally, the TRPN subfamily was given the name no mechanoreceptor potential C (nompC) [10]. The TRPA family specializes in temperature detection and is associated with the activation of pain detection pathways, causing avoidance behavior and promoting inflammatory responses [9]. The TRPN family has activation mechanisms associated with mechanosensation [1]. Currently, the most important issue is that research on TRP channels has become a major target with great relevance to the discovery and development of new drugs and provides key clues for the treatment of various diseases [11]. Many studies have reported that the various physiological regulatory functions of TRP channels are related to numerous diseases and could serve as potential treatments. The ongoing research on TRP channels has revealed their importance to the scientific community. Consequently, in 2021, David Julius and Ardem Patapoutian were awarded the Nobel Prize in Physiology/Medicine for their research on the discovery of temperature-sensitive TRP channels and pressure-sensitive piezo channels [12].

In this review, I have focused on the TRPA1 channel and TRPA1-related diseases among the TRPA family. TRPA1 functions as a non-selective cationic channel expressed on various types of cell membranes. This channel is involved in the sensory transmission of specific odors, tastes, and chemicals, and, similar to other TRPA family channels, it regulates responses to temperature and many other stimuli [9]. Recent studies have demonstrated the close association between TRPA1 and various disorders and diseases. For example, TRPA1 performs a critical function in the chronic pain control mechanism and has been shown to be deeply related to inflammatory diseases. The activation of TPRA1 can cause pain, which could be a key factor in the study of chronic pain diseases [13]. Additionally, mutations in TRPA1 can affect sensory nerves and microvasculature, which may be associated with neuralgia and vascular diseases [14]. For example, TRPA1 activation is considered important in the immune response, so it may be closely related to allergies and other immune-related diseases [15]. Additionally, TRPA1 activation associated with particulate matter, a rapidly growing problem, may have implications for respiratory diseases [16]. In particular, exposure to environmental toxins can worsen bronchospasms and other respiratory symptoms [17]. Therefore, understanding the function and regulatory mechanisms of TRPA1 should be a top priority to find effective approaches for the appropriate treatment and prevention of various diseases. Further research is underway in this area, and understanding the exact relationship between TRPA1 and the disease will help in the development of potential treatment strategies.

## 2. TRPA1 Structure

TRPA1, like other TRP channels, is composed of four homomers and has six transmembrane alpha helices from S1 to S6; a single pore is structurally located in the hydrophilic region between S5 and S6 [18]. Sixteen ankyrin repeats were observed at the N-terminus of TRPA1, which forms a convex stem structure consisting of five ankyrin repeats and 11 additional ankyrin repeats. These ankyrin repeats are structurally important for the chemical and thermal sensitivity of the channel as well as its interactions with cytoskeletal proteins [19,20]. Three cysteines (C621, C641, and C665) and one lysine (K710) present at the proximal N-terminus are covalently modified by reactive electrophiles, a mechanism essential for TRPA1 activation by electrophiles [21]. Representative examples of these reactive electrophiles include allyl isothiocyanate (AITC), cinnamaldehyde, cuminaldehyde, anisaldehyde, and tiglic aldehyde [22,23]. Additionally, activation of TRPA1 by non-electrophiles occurs primarily through non-covalent binding. In particular, non-electrophiles activate ion channels by non-covalently binding to the S5/S6 domains of TRPA1, which form a part of the transmembrane and are closely involved in non-electrophilic TRPA1 activation [24]. Menthol, carvacrol, clotrimazole, thymol, decanol, 2-ethyl-1-hexanol, toluene, and α-terpineol are well-known non-electrochemicals involved in TRPA1 activation [25]. This non-electrophilic activation of TRPA1 enables chemosenses associated with the perception of temperature, texture, and pungency [26] (Figure 1).

## 3. Physiological Roles of TRPA1

TRPA1 acts as a cellular sensor that transmits sensations of pain, itching, and temperature through the peripheral and central regions of the brain.

### 3.1. TRPA1’s Role in Pain

In the somatosensory system, nociceptors are distributed on primary afferent nerve fibers, and their activation causes pain by detecting chemical, mechanical, and thermal stimulations [27,28]. Through studies on cutaneous nerve fibers, activation of TRPA1 by chemical factors has been identified to cause neurogenic axonal reflex erythema, acanthosis, cold hypersensitivity, mechanical hyperalgesia, heat sensation, and pain [29,30,31]. TRPA1 is constitutively activated in response to endogenous agonists [32]. These mechanisms demonstrate that TRPA1 is closely related to chronic pain in various diseases, such as emphysema, bronchitis, fibromyalgia, diabetes, migraine, osteoarthritis, inflammation, and neuropathy [18,33,34]. Therefore, research on TRPA1 could be a suitable solution for discovering new treatments for inflammation and pain [35,36,37].

### 3.2. TRPA1’s Role in Itching

TRPA1, a nociceptor found in various skin components, is closely related to histamine-independent itching. Most cases of chronic itching are histamine-independent, and histamine plays a minor role [38]. TRPA1 is activated by cold temperatures and various itch-causing substances (both endogenous and exogenous) and specifically mediates itching induced by chloroquine, cowhage, and many other compounds [39,40]. TRPA1’s involvement in itching has been clearly shown in animal studies and human skin biopsies, especially in diseases such as allergic dermatitis and atopy [41]. TRPA1’s interactions with thymic stromal lymphopoietin and interleukin-31 (IL-31) are associated with itching [42,43]. Inflammatory mediators, bile acids, and reactive oxygen species are also major factors that activate TRPA1 and cause itching [44]. TRPA1 has great potential as a target for alleviating cholestatic pruritus and other pruritic symptoms. TRPA1 also influences epidermal barrier recovery and responds to several stimuli, such as activation of Toll-like receptor 7, endothelin, and periostin [44]. Understanding the multifaceted involvement of TRPA1 offers potential solutions for the treatment of chronic itching.

### 3.3. TRPA1’s Role in Temperature

TRPA1 showed increased activation when the temperature decreased to less than 18 °C in a study using mouse DRG neurons and CHO cells. Based on this evidence, TRPA1 was initially accepted as an ion channel that mainly plays a role in detecting low temperatures [45,46]. However, whether mammalian TRPA1 can detect heat remains unclear [47]. TRPA1 in neurons appeared to sense noxious heat, and experiments performed on knockout mice confirmed heat-induced TRPA1 activation [48]. Additionally, TRPA1 activation by sensing cold and hot temperatures may occur through a pathway activated by reactive oxygen species (ROS) generated by rapid temperature changes [49].

## 4. Diseases Associated with TRPA1

TRPA1 is significantly involved in pain, temperature sensation, and cellular function and is known to have high Ca^2+^ permeability among non-selective ion channels. Several studies have implicated TRPA1 in diseases caused by abnormal functions of various cells, including genetic mutations in the channel. Therefore, understanding the functions of TRPA1 in relevant diseases is of utmost importance.

### 4.1. Alzheimer’s Disease (AD) and TRPA1

AD is a progressive neurodegenerative disease that is a well-known cause of senile dementia and is highly related to TRPA1 [50]. Dr. Alzheimer, the German psychiatrist who first introduced the disease, observed characteristic brain tissue findings, such as neurofibrillary tangles and amyloid plaques, under a microscope in postmortem brain tissue from patients with delusions and cognitive impairment. This can result in selective nerve loss, inflammation, and severe oxidative damage. Therefore, these symptoms can cause dementia [51]. Astrocytes are responsible for maintaining central nervous system (CNS) function and homeostasis through ion transport, neurotransmitter absorption, and neurotrophic factor production and have a very high CNS distribution [52]. A previous study identified widespread expression of TRPA1 in astrocytes distributed in the trigeminal occipital nucleus of rats [53]. Inflammatory factors, nitric oxide, and ROS open TRPA1 channels in astrocytes and initiate Ca^2+^ influx, which is the main mechanism that activates PP2B signaling and inflammatory response in astrocytes [54,55]. An imbalance in Ca^2+^ concentration results in excessive activation of astrocytes, releasing inflammatory factors that lead to neurodegeneration. NFAT, PP2B, and NF-κB are activated and trigger the inflammatory response responsible for the pathogenesis of AD [55]. In an AD mouse model, a TRPA1 blockade promoted the maintenance of structural synaptic integrity and normalization of neuronal function and astrocyte activity [51]. This suggests that AD progression can be slowed by inhibiting the TRPA1 channel function. In short, the presence or absence of TRPA1 channel activation has opposite effects on astrocytes. The process by which TRPA1 introduces Ca^2+^ and transmits signals to PP2B may be essential for regulating astrocyte-related inflammation and TRPA1 inactivation promotes neuroprotection. Therefore, it has emerged as an effective therapeutic option. Recent studies have implicated the TRPA1 calcium channel in astrocytes in the early toxicity of AD, and blocking TRPA1 has been shown to reverse abnormal astrocyte and neuronal activity. TRPA1 can be inhibited using HC030031, a specific inhibitor that has shown efficacy in the treatment of neuropathic pain and can safely cross the blood-brain barrier [51]. In another study, memantine (MEM) treatment reduced apoptosis, mitochondrial oxidative stress, expression, and intracellular Ca^2+^ concentration [56]. When TRPA1, which has a high Ca^2+^ affinity is activated, intracellular ROS production increases excessively owing to abnormally high intracellular Ca^2+^ concentrations. This mechanism is one of the main causes of neurodegenerative diseases such as dementia and AD. MEM, used to treat AD, indirectly induces inhibition of TRPA1 [56]. This reduces intracellular Ca^2+^ concentration and ROS generation, and this interaction has significant implications in not only AD but also neurological diseases such as memory loss and aging. Melatonin has been shown to exert significant neuroprotective effects in AD-related studies [57]. These effects of melatonin are mediated by TRPA1, which is consistent with the data from recent studies [58]. Additionally, studies have shown that sleep-related disorders are significantly associated with the development of AD [59]. Therefore, melatonin is involved in both sleep disturbance and AD-related cell death and may be considered as a potential therapeutic option for preventing dementia in patients with AD.

### 4.2. Migraine and TRPA1

Migraines are mainly characterized by throbbing or pounding pain on one side of the head or temporal area and are different from tension-type headaches, in which the entire head feels stiff and tight. Severe symptoms can interfere with daily life, and academic and work abilities are often impaired. Some people experience prodromal symptoms before a migraine, such as sensitivity to sound or light, stiff neck, difficulty concentrating, and fatigue. In some cases, cases this symptom appears after the pain disappears. A recent study has shown that TRPA1 is closely involved in the pathogenesis of migraines [60]. Inflammation and oxidative stress in brain blood vessels may be important causes of migraines. TRPA1 may regulate blood flow in brain vessels in response to inflammatory and oxidative stress, which may contribute to migraine development [61]. Additionally, changes in cerebral blood flow are observed when migraines occur, and TRPA1 has been hypothesized to contribute to the regulation of these changes in cerebral blood flow [62]. Headache, one of the main symptoms of migraines, can appear in the form of pain; the pain detection function, a key feature of TRPA1, is noteworthy in the signaling pathway related to the occurrence of these headaches [63]. Recently, the importance of endothelial TRPA1 in regulating cerebral blood flow through neurovascular coupling was demonstrated, particularly as a sensor of cellular redox signaling associated with cerebrovascular diseases [64]. Endothelial TRPA1, which is present in various cerebral blood vessels, immediately responds to hypoxia and cerebral ischemia by dilating the epithelial arteries, parenchymal arterioles, and capillaries. Previous studies have shown that increased endothelial TRPA1 activity can reduce cerebral infarct size. Additionally, capillary endothelial TRPA1 promotes neurovascular coupling by inducing the dilatation of upstream parenchymal arterioles [62]. In sensory neurons, TRPA1 activation results in the release of calcitonin gene-related peptide (CGRP), which acts as a vasodilator. Neuronal TRPA1 activation may cause dilatation of the cerebral vasculature, leading to headaches [45,65]. In summary, endothelial TRPA1 activation exerts a protective effect on neurons by regulating cerebral blood flow, but the activation of neuronal TRPA1 can cause harmful brain symptoms and contribute to conditions such as headaches and age-related cognitive decline. Parthenolide is a naturally occurring compound found in certain plants such as the feverfew (*Tanacetum parthenium*) and other family members, and it has been found to have analgesic, anti-cancer, and anti-inflammatory effects [66]. Parthenolide is a partial TRPA1 agonist that targets TRPA1, desensitizes TRPA1 channels, and inhibits TRPA1-mediated pain responses. This inhibitory mechanism is the principle of selectively desensitizing TRPA1-mediated pain while suppressing neuropeptide-mediated neurogenic responses by potentially disabling the function of TRPA1-expressing neurons [67]. Parthenolide, which inhibits TRPA1-mediated pain responses, has anti-inflammatory effects and prevents migraines. *N*-acetyl-parabenzoquinone-imine, a metabolite of acetaminophen, acts on TRPA1 to produce a mild neurogenic inflammatory response but also activates spinal TRPA1, resulting in the long-term inhibition of voltage-gated sodium and calcium currents in primary sensory neurons [68]. Pyrazolone derivatives, such as propyphenazone and dipyrone, selectively inhibit TRPA1, thereby alleviating migraines independent of prostaglandin production [69].

### 4.3. Asthma and TRPA1

Asthma is caused by a variety of environmental and genetic factors, and it is a chronic airway disease that occurs in the bronchi of the lungs. The bronchi become sensitive and narrow, causing symptoms such as difficulty in breathing, chest tightness, wheezing sounds (wheezing) when breathing, and coughing. These symptoms tend to worsen at night, early in the morning, or when the patient has a cold. Asthma symptoms are characterized by repeated worsening and improvement; however, inflammation persists, even in the absence of symptoms. If the inflammatory condition is left untreated, the structure of the bronchial tubes will be deformed and lung function will decline, causing breathing difficulties that appear with age. The introduction of biological agents has significantly improved asthma management, especially for type 2 asthma, by targeting cytokine-signaling pathways. However, further elucidation of the pathological mechanisms is required for subsets of patients who do not respond well to current treatments, including those with type 1 or mixed type 1/2 asthma [70]. Recent research suggests that TRPA1, a stimulus-activated channel found in the sensory nerves of the lungs, is critical for regulating lung inflammation and airway hyperresponsiveness [71]. A previous study investigated the association between asthma and TRPA1. In this study, mice with normal expression of TRPA1 and asthmatic mice in which TRPA1 was genetically deleted or pharmacologically inhibited were used. The results showed a close association between TRPA1 and various inflammatory markers such as IL-4 and IL-13, PGD2, and NGF, which are important for asthma development [72]. To confirm the relationship between TRPA1 and the pro-inflammatory response observed in asthmatic mice, lung tissue from an asthmatic mouse model was analyzed and the expression and mRNA levels of TRPA1 were increased to the levels of inflammatory markers. However, in mice in which TRPA1 was deleted or pharmacologically inhibited, airway inflammation was reduced and no significant changes in inflammatory markers were found, suggesting that TRPA1 is a potential factor that mediates airway inflammation. Additionally, according to this study, TRPA1 is involved in these neuroimmune interactions, specifically through TRPA1 expression on CD4^+^ T cells, which could potentially affect immune cell activation and inflammatory responses. Moreover, studies on asthma exacerbation caused by factors such as PM2.5 pollution suggest that TRPA1 may contribute to enhancing airway inflammation associated with environmental pollutants, further solidifying the involvement of TRPA1 in asthma pathogenesis [72]. In summary, TRPA1 may serve as a promising therapeutic candidate protein among asthma treatment strategies. However, more research is required to identify the involvement of TRPA1 signaling pathway in immune cell function and airway inflammation in asthma.

### 4.4. Diabetes Mellitus (DM) and TRPA1

DM is a disease in which blood sugar levels increase above normal levels. It is a metabolic disease caused by insufficient insulin secretion or glucose that is not absorbed by the body cells. Therefore, it is characterized by high blood sugar, which causes various symptoms, and excretion of glucose through urine. Type 1 DM (T1DM) occurs when a genetic defect causes insulin secretion from pancreatic beta cells to decrease to less than normal. Although the exact cause is unknown, type 2 DM (T2DM) develops due to increased insulin resistance. Most individuals with T2DM develop insulin resistance in adulthood for various reasons. T1DM develops during childhood because of insufficient insulin secretion from the pancreas. However, due to other exceptions, it is advisable to diagnose T1DM and T2DM by measuring insulin secretion and resistance. Obesity is a major cause of T2DM in many cases [73]. Weight gain increases free fatty acid concentration, ultimately disrupting glucose homeostasis and causing insulin resistance in various tissues, leading to the development of T2DM [74]. Dysfunctional white adipose tissue in obesity exacerbates insulin resistance and promotes T2DM development by storing excess lipids around vital organs [75]. TRPA1, a channel that has been extensively studied for its association with glucagon-like peptide-1 (GLP-1) and insulin secretion, is suitable as an effective therapeutic target for dietary insulin resistance and associated diseases [36]. TRPA1 activation by various compounds stimulates the insulin secretory mechanism in pancreatic beta cells. Additionally, Roux-en-Y gastric bypass surgery is involved in regulating TRPA1 expression and contributing to improved glucose homeostasis in diabetic rats [76]. Estrogen metabolites and certain dietary compounds may also activate TRPA1 to promote insulin secretion and GLP-1 release from intestinal L cells, which may be an appropriate method for diabetes management [77]. In addition, AITC and cinnamaldehyde interact directly with TRPA1 to improve insulin secretion and weaken insulin resistance, thereby showing anti-diabetic effects. Activation of AMP-activated protein kinase (AMPK) downregulates TRPA1 expression. These processes suggest that AMPK regulates TRPA1-mediated responses [78,79]. In conclusion, controlling TRPA1 activation has the potential to combat obesity and T2DM by regulating insulin secretion, GLP-1 release, and insulin sensitivity [80]. Additional studies are required to develop effective therapeutic strategies that leverage the potential of TRPA1 in the management of DM.

### 4.5. Cancer and TRPA1

Breast cancer is a malignant tumor that occurs in the breast and can be life-threatening if it metastasizes to other organs outside the breast. The breast is composed of various types of cells, and any cell can turn into a cancer cell; thus, many types of breast cancer can develop. Among the symptoms of breast cancer, chronic pain significantly reduces patients’ quality of life. Opioids, which are commonly used to manage cancer pain, have side effects [81]. Recent studies focused on TRPA1, which is implicated in neuropathic pain, and analyzed alternative mechanisms to alleviate this pain [82]. Experimental results have shown that treatment with TRPA1 antagonists can effectively manage neuropathic pain caused by chemotherapy and cancer pain [83]. Experiments using mice with metastatic melanoma and breast cancer have shown that TRPA1 activation induces voluntary pain behavior [84]. Furthermore, studies using genetic and pharmacological approaches to determine the role of TRPA1 in a murine breast carcinoma model have shown that injections of TRPA1 antagonists and antisense oligonucleotides reduce spontaneous pain-related behaviors and mechanical and cold hypersensitivity [85]. Additionally, activation and sensitization of TRPA1 were observed in response to increased H_2_O_2_ levels. In summary, TRPA1 senses reactive compounds such as H_2_O_2_ and acts as a cancer pain modulator [86].

Melanoma occurs when melanocytes that produce melanin pigment become malignant. Therefore, it can occur anywhere where melanocytes are found. However, it most often occurs on the skin and is the most malignant skin cancer. Early in melanoma progression, proinflammatory M1 macrophages contribute to the mutagenic microenvironment [87]. However, in advanced melanoma, tumor-associated macrophages (TAMs) shift toward an anti-inflammatory M2 phenotype, promoting tumor growth and immunosuppression [87]. High numbers of TAMs, especially CD163+ve TAMs, are correlated with tumor invasion, metastasis, and poor clinical outcome [88]. TRPA1 activation in melanoma cells involves canonical ROS-neutralization mechanisms and non-canonical oxidative stress defense, leading to cancer progression through anti-apoptotic pathways [89]. TRPA1, which is highly sensitive to oxidants, is functionally expressed in melanoma cells and can be activated by H_2_O_2_ to produce calcium-dependent responses. Therefore, activation of TRPA1 by H_2_O_2_ may promote an anti-apoptotic, pro-oncogenic program in melanoma [86]. Importantly, TRPA1 activation in melanoma cells amplifies oxidative stress signaling. These findings demonstrate the complex relationships between TRPA1, macrophages, and oxidative stress throughout the progression of melanoma, and further in-depth research in this field is needed.

### 4.6. Renal Failure and TRPA1

The kidneys filter unnecessary waste products, such as urea and nitrogen, from the blood pumped out by the heart. Even in emergency situations, due to environmental changes, the kidneys always maintain an appropriate amount of moisture in the body and a constant blood concentration. In this way, when the kidneys are temporarily damaged and return to normal only after a certain period, it is called “acute renal failure.” In contrast, a phenomenon in which kidney tissue is permanently damaged and kidney function declines for more than 3 months is called “chronic renal failure” [90]. Recent studies have implicated TRPA1 in kidney disease and primarily addressed its effects on non-neuronal cells [91]. During kidney damage caused by sepsis, TRPA1 has been shown to function as a protective mechanism, regulate mitochondrial dynamics, and reduce inflammation and oxidative stress [92]. Additionally, TRPA1 alleviates renal ischemia-reperfusion injury and angiotensin II-induced renal injury by suppressing macrophage-mediated inflammation. Increased TRPA1 expression in the renal tubules is closely related to poor renal function recovery and the severity of acute tubular necrosis, suggesting that TRPA1 is a potential risk factor [93]. Moreover, TRPA1 activation in renal tubular cells exacerbates injury and inflammation, whereas the absence of TRPA1 attenuates these effects [93]. Overall, TRPA1 appears to have diverse effects on renal epithelium and macrophages, suggesting a multifaceted role in renal inflammation and disease. Further studies using cell-specific TRPA1 deletion models are required to elucidate its functional significance in kidney diseases.

### 4.7. Post-Operative Ileus (POI) and TRPA1

POI is a common complication of gastrointestinal adhesion and stagnation of digestive tract movement due to surgery. A recent study has proven the effectiveness of regulating the functions of TRPA1 and TRPV1 by using natural drugs as a treatment method for POI [94]. Natural medicines extracted from plants and herbs are attracting attention because of their biocompatibility and safety, and they relieve clinical pain and improve digestive function by controlling calcium influx and cholecystokinin secretion or serotonin release by regulating TRPV1/TRPA1 activity [95]. Overall, targeting TRPV1/TRPA1 has the potential to treat POI, but the efficacy and safety of the treatment need to be evaluated. Another study showed that early oral nutrition alleviates the degranulation observed in POI, in which mast cells become activated and release substances contained in intracytoplasmic granules [96]. High-fat enteral nutrition has been associated with reduced levels of certain inflammatory factors in POI, potentially contributing to mast cell stabilization; early oral feeding may stimulate splanchnic nerves, thereby affecting mast cell function [97]. Additionally, the expression of TRPA1, which is involved in POI, was enhanced by early oral feeding. TRPA1 plays a role in regulating gastrointestinal transport, and its activation was associated with improved intestinal function in a POI model [98]. TRPA1 activation controls mast cell degranulation by increased cholecystokinin 1 receptor (CCK1-R) expression [96]. Furthermore, TRPA1 and CCK1-R appear to be involved in communication between mast cells and neurons [96]. Overall, early oral nutrition may be a potential treatment for POI because it regulates mast cell activity via the TRPA1/CCK1-R pathway.

### 4.8. Coronavirus Disease (COVID-19) and TRPA1

The COVID-19 pandemic was a great disaster for humanity that spread due to SARS-CoV-2 virus infection at the end of 2019. SARS-CoV-2 infection generally starts with a mild-to-moderate respiratory illness and recovery occurs without special treatment in approximately a week to 15 days. However, some cases can worsen, requiring treatment and even leading to death. Although COVID-19 incidence rates are difficult to compare across countries owing to the lack of standardized assessment methods, mortality rates can be used as an indicator of COVID-19 severity, and they vary greatly across countries [99,100]. In regions with low mortality rates, such as Africa, the Balkans, Central Europe, and East Asia, high consumption of fermented vegetables [101] and, in other cases, consumption of spices has been observed. Recent studies have reported that the COVID-19 mortality rate decreases as spice consumption increases [102], and high consumption of both spices and fermented vegetables has been reported in these countries [103,104]. Oxidative stress storm is the most representative characteristic of the clinical symptoms of COVID-19 [105]. Consumption of fermented vegetables can activate nuclear factor erythroid 2-related factor 2 (Nrf2) [101,106,107], which produces a variety of Nrf2-interacting nutrients (sulforaphane, resveratrol, quercetin, genistein, gallate, epigallocatechin, curcumin, and berberine) that help reduce cytokine storm, lung damage, endothelial damage, and insulin resistance [101]. Recent studies have shown that foods and nutrients that interact with Nrf2 rebalance insulin resistance and are closely associated with COVID-19 severity [101,108,109,110]. This interaction between Nrf2 and TRPA1 suggests that Nrf2 significantly affects TRPA1 via ROS. Pulmonary ischemia-reperfusion injury increases oxidative stress in the brainstem by interfering with the antioxidant responses of Nrf2 and TRPA1 via ROS [111]. Additionally, brain polysulfides activate TRPA1 and enhance Nrf2 translocation [112]. TRPA1 knockdown in mice aggravates inflammation and kidney damage after ischemia-reperfusion injury [93]. Several animal studies have shown that neuroprotection occurs when the Nrf2 pathway is activated through antioxidant signaling [113,114,115,116]. Additionally, redox-sensitive calcium channels overexpressed in human cancers enhance ROS resistance by activating calcium-dependent anti-apoptotic pathways. Nrf2 directly regulates TRPA1 expression, providing a complementary protective mechanism against oxidative stress along with traditional ROS neutralization strategies [89].

## 5. Application of Nanomedicine to Ion Channels-Related Diseases

Recent developments in nanomedicine targeting ion channels have shown a significant impact on the treatment of various diseases and are emerging as an innovative alternative by effectively overcoming various side effects and shortcomings of existing treatment methods. Nano-delivery systems (NDS) are a representative nanomedicine technology that maximizes treatment effects by delivering drugs or genetic materials to precise locations using nano-sized particles. Additionally, NDS is designed to improve drug stability, biocompatibility, and efficient delivery to the target site.

### Nanoparticles Used in the Latest NDS for Disease Treatment Targeting Ion Channels

The first ais ligand-based targeting nanoparticles. Their selectivity is increased by attaching a peptide that targets a specific ion channel to the surface of a nanoparticle, through which the nanoparticle selectively reaches the diseased area and releases the drug. This type of NDS is expected to exert a neuroprotective effect by suppressing the abnormal activity of ion channels. The second ispH-responsive nanoparticles, which use the principle of being activated in the specific pH environment of the diseased area and releasing the drug, making them effective not only for tumors but also for inflammatory areas. The third is magnetically guided nanoparticles. Using an external magnetic field, the nanoparticles are guided to a specific area and allowed to bind to the ion channel. Abnormal blood flow can be improved using a magnetic field to guide magnetically induced nanoparticles to the site of cardiovascular disease, targeting ion channels to deliver drugs into blood vessels and controlling ion channels in blood vessel lining cells. The fourth is immune nanoparticles. Similar to the first ligand-based targeting nanoparticle, they regulate the immune response by attaching antibodies or ligands targeting immune cells to the nanoparticle to regulate specific ion channels. Immune nanoparticles can reduce inflammation and alleviate disease symptoms by binding to ion channels on the surface of immune cells and suppressing or regulating autoimmune responses. The last are gene-editing nanoparticles. The CRISPR/Cas9 system can be mounted on the nanoparticles to edit specific ion-channel genes and can be used to treat genetic diseases.

## 6. Application of Nanoparticles in TRPA1-Related Diseases

### 6.1. Treatment Methods to Inhibit TRPA1

#### 6.1.1. Periocular Mechanical Allodynia (PMA)

The calcitonin receptor-like receptor/receptor activity-modifying protein 1 (CLR/RAMP1) complex in Schwann cells is activated by CGRP and induces periocular mechanical allodynia (PMA). This activation occurs in endosomes within Schwann cells and initiates the pain pathway. CLR/RAMP1 in Schwann cell endosomes signals through cAMP-dependent nitric oxide synthase (NOS) activation, generating nitric oxide (NO). NO then mediates pain by targeting TRPA1 and NOX1 to generate ROS, which sustains ROS production, targets nociceptors, and results in the transmission of allodynic signals to the CNS [117]. Administration of the nanoparticle-encapsulated antagonist DIPMA-MK-3207, a pH-responsive nanoparticle targeting endosomal CLR/RAMP1 [118], was found to improve CGRP-induced pain-relieving efficacy by blocking TRPA1-involved signaling (Figure 2a).

#### 6.1.2. Breast Cancer

Cancer cells use an oxidative stress defense system to suppress ROS accumulation [119]. Breast cancer cells overexpressing the TRPA1 ion channel exhibit resistance to conventional chemotherapeutic drugs owing to the activation of anti-apoptotic pathways. To overcome the limitations of these treatments, a new treatment method using photothermally conjugated polymer nanoparticles (NPs-H), which encapsulate HC-030031 (HC), a specific inhibitor of the TRPA1 ion channel that responds to near-infrared (NIR) light, was introduced [120]. When these nanoparticles are exposed to an 808 nm laser, they block Ca^2+^/CaM complex formation, reducing chemotherapy resistance and promoting tumor cell death [120]. Nanoparticles that inhibit TRPA1 show significant potential as cancer treatments, especially for cancers overexpressing TRPA1, thus providing a promising approach for overcoming existing chemotherapy resistance (Figure 2b).

#### 6.1.3. Multiple Sclerosis (MS)

MS, a chronic inflammatory and autoimmune disease of the CNS, is characterized by increased ROS [121,122,123]. Inflammation significantly contributes to the development of MS [124]. Among the TRP channels, TRPA1 is known to be involved in inflammation and immune responses and is expressed in various CNS tissues to regulate intracellular calcium and apoptosis. Inhibition of TRPA1 can prevent demyelination, making TRPA1 an effective target for MS treatment. A critical drawback of ivermectin (IVM), an anthelmintic drug with anti-inflammatory properties, is its low bioavailability, but solid lipid nanoparticles (SLN) have the advantage of increasing the bioavailability of hydrophobic drugs such as IVM [125]. SLN, composed of solid lipids and surfactants, are able to improve drug stability and delivery and could be an effective treatment for MS that specifically acts on the TRPA1/NF-kB/GFAP signaling pathway (Figure 3a).

#### 6.1.4. Knee Osteoarthritis (KOA)

KOA is a widespread degenerative joint disease worldwide, characterized by joint pain, swelling, and stiffness [126]. Synovitis is an important factor in the development of KOA, with TRPA1 amplifying inflammation in fibroblast-like synoviocytes [127,128]. Recent studies have reported decreased activity of AMPK, which negatively regulates TRPA1 in KOA [129]. Therefore, essential oil was extracted from Sanse Powder, a herbal medicine for KOA, and a nanoemulsion (SP-NE) with excellent stability was manufactured through a high-pressure homogenization process [130,131]. In vitro and in vivo studies using SP-NE have shown that the oil alleviates KOA synovitis by negatively regulating TRPA1 through AMPK-mTOR signaling [132]. These results serve as a stepping stone for new treatments using nanotechnology for TRPA1-related diseases (Figure 3b).

### 6.2. Treatment Methods to Activate TRPA1

#### 6.2.1. Tumor

Tumors pose a serious health risk and require effective treatment [133]. Ion interference therapy, which reverses the distribution of ions in tumor cells using bioactive nanomaterials, has attracted attention because of its safety and low risk of drug resistance [134]. Ca^2+^ plays a pivotal role in cellular processes, and calcium overload can induce cell death [135]. Currently, traditional Ca^2+^ nanogenerators have serious drawbacks owing to stability and reactivity issues [136]. Therefore, to address these limitations of exogenous Ca^2+^ nanogenerators, mesoporous membranes loaded with indocyanine green and NO donors and coated with hyaluronic acid were developed with the idea of using dual sources of endogenous Ca^2+^. Silica nanoparticles (MSNs), I/B@MSN-T@HA (light-controlled Ca^2+^ nanomodulators), have been developed and possess tumor cell-targeting properties [137]. NIR light irradiation triggers ROS generation and Ca^2+^ influx through TRPA1 channels, whereas NO release opens ER channels and causes Ca^2+^ leakage. This dual-channel approach maximizes intracellular Ca^2+^ concentration, leading to mitochondrial dysfunction, ATP reduction, and tumor cell death. This strategy has been shown to effectively inhibit tumor growth by inducing Ca^2+^ overload and leveraging the body’s intrinsic Ca^2+^ regulatory system without external ion influx [137]. This approach minimizes the side effects and can potentially be applied to other ionic therapies (Figure 4).

#### 6.2.2. Diabetes

Optogenetics uses light to precisely control cellular activity, but its use is limited by poor tissue penetration and the need for invasive procedures. NIR light offers a non-invasive alternative with greater tissue penetration [138,139,140,141,142,143]. Additionally, despite the difficult preparation steps and unstable photostability, NIR-absorbing nanomaterials, especially organic composites, have superior potential because of their optoelectronic properties and light-harvesting abilities [144,145,146,147]. In a recent study, biomimetic conjugated oligomeric nanoparticles (CM-CON) with NIR-II imaging capabilities, ROS generation, and photothermal effects were developed using an “acceptor-donor” conjugated oligomer (S81CF) encapsulated in endocrine cell membranes [148]. When exposed to an 808 nm laser, CM-CON generates ROS and local heat, which activates TRPA1 and TRPV1 ion channels to mediate Ca^2+^ influx and increase GLP-1 secretion, thereby lowering blood glucose levels [148]. CM-CON improved glucose homeostasis, reduced obesity, showed good biosafety, and inhibited liver and islet damage in db/db mice [148]. This study presents a promising NIR-based strategy for glucose control and provides important insights for diabetes treatment (Figure 4).

## 7. Conclusions

TRPA1, a non-selective cation channel, is involved in the sensory transmission of specific odors, tastes, and chemicals and temperature regulation. This channel is involved in important physiological functions in the human body, such as pain sensation, itching, and temperature detection. Recent studies have revealed its important functions in chronic pain mechanisms and its association with inflammatory diseases, neuralgia, vascular diseases, immune responses, and respiratory diseases, suggesting that TRPA1 may be an important target for the treatment of several diseases.

Understanding the structural and functional dynamics of TRPA1 is essential for the development of targeted treatments for related diseases. Structural elements of the channel, including the transmembrane α-helix, ankyrin repeats, and activation mechanisms by electrophilic and non-electrophilic substances, can be an important guide for understanding its operation and regulation.

TRPA1 is involved in various diseases due to its role in pain, temperature sensation, and cellular function, especially through high Ca^2+^ permeability. In AD, TRPA1 activation in astrocytes promotes inflammation and oxidative stress, exacerbating neurodegeneration, whereas TRPA1 inhibition enhances neuroprotection. In migraine, TRPA1 regulates cerebral blood flow and pain, suggesting that the blockade of TRPA1 may be a therapeutic approach. In asthma, TRPA1 mediates airway inflammation and hyper-responsiveness, making it a potential therapeutic target. In diabetes, TRPA1 regulates insulin secretion and sensitivity, indicating its therapeutic potential. In cancers, especially breast cancer and melanoma, TRPA1 affects pain and tumor progression through oxidative stress responses. In renal failure, the role of TRPA1 in inflammation and injury suggests that it is involved in the mechanisms of kidney disease. Additionally, TRPA1 has been implicated in POI through the regulation of gastrointestinal motility, and its interaction with Nrf2 in COVID-19 influences oxidative stress and inflammation, indicating its broad significance in disease pathophysiology and potential therapeutic interventions.

Additionally, the application of nanomedicines to target TRPA1-related diseases represents a promising method for improving treatment efficacy and reducing side effects. Technologies such as ligand-based targeting, pH-responsive nanoparticles, magnetically guided nanoparticles, immune nanoparticles, and gene-editing nanoparticles are becoming a solid foundation for innovative approaches to precise and effective therapeutic interventions.

In summary, continued research on TRPA1 is essential to unlock its full potential for disease treatment and prevention. As our understanding deepens, TRPA1 may become the cornerstone for the development of new therapeutic strategies, ultimately improving patient outcomes in various chronic and acute diseases.

## Figures and Tables

**Figure 1 ijms-25-09234-f001:**
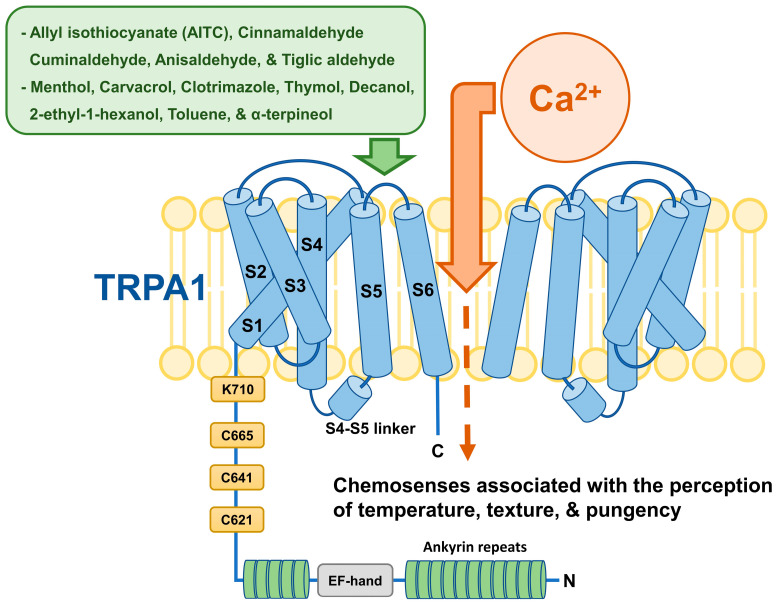
TRPA1 has four subunits and six transmembrane helices. It has 16 ankyrin repeats at the N-terminus important for sensitivity to chemicals and temperature. Three cysteines (C621, C641, and C665) and one lysine (K710) are modified by electrophiles (e.g., AITC, cinnamaldehyde, cuminaldehyde, anisaldehyde, and tiglic aldehyde) to activate TRPA1. Non-electrophiles (e.g., menthol, carvacrol, clotrimazole, thymol, decanol, 2-ethyl-1-hexanol, toluene, and α-terpineol) activate it by binding to the S5/S6 domains. This activation mode is crucial for sensing temperature, texture, and pungency. Green arrow: TRPA1 activation, orange arrow: external calcium ion influx due to TRPA1 activation.

**Figure 2 ijms-25-09234-f002:**
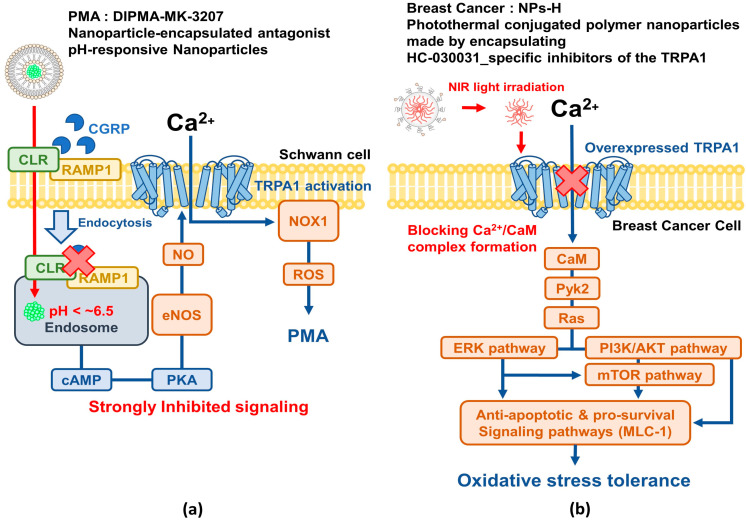
Application of nanotechnology in breast cancer and PMA treatment targeting TRPA1 inhibition. (**a**) Nanoparticle-encapsulated antagonist DIPMA-MK-3207, a pH-responsive nanoparticle targeting endosomal CLR/RAMP1, enhances CGRP-induced pain relief efficacy by blocking TRPA1-related signaling.; (**b**) photothermal conjugated polymer nanoparticles (NPs-H) encapsulating HC-030031 (HC), a specific inhibitor of the near-infrared (NIR)-responsive TRPA1 ion channel, block the formation of Ca^2+^/CaM complexes when exposed to an 808 nm laser. As a result, chemotherapy resistance is reduced and tumor cell death is promoted. Red arrows: Pathway of disease suppression by nanoparticles, blue arrows: Pathway of disease development by TRPA1 activation.

**Figure 3 ijms-25-09234-f003:**
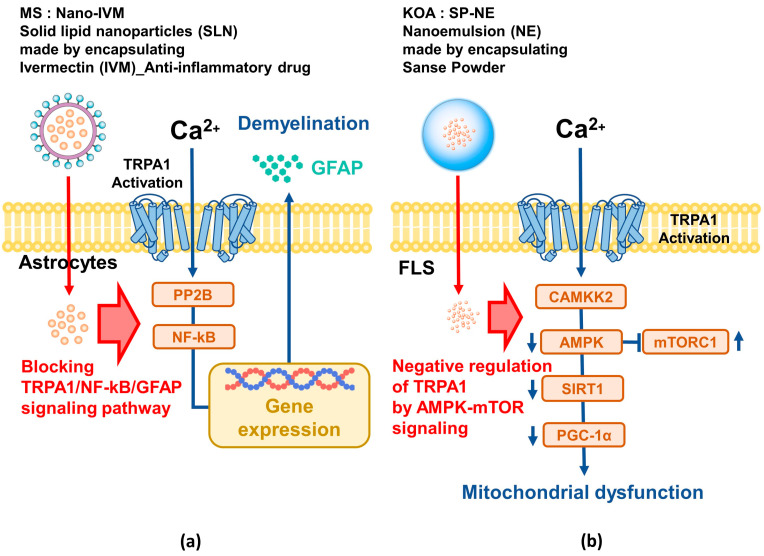
Application of nanotechnology in MS and KOA treatment targeting TRPA1 inhibition. (**a**) The use of solid lipid nanoparticles (SLNs) to deliver anti-inflammatory drugs with poor bioavailability could improve drug stability and delivery, potentially enabling effective MS treatment to prevent demyelination by inhibiting the TRPA1/NF-kB/GFAP signaling pathway.; (**b**) Nanoemulsion (SP-NE) with excellent stability, manufactured by extracting essential oil from pickling powder, a herbal medicine for KOA, through a high-pressure homogenization process, alleviates KOA synovitis by negatively regulating TRPA1 through AMPK-mTOR signaling. Red arrows: Pathway of disease suppression by nanoparticles, blue arrows: Pathway of disease development by TRPA1 activation.

**Figure 4 ijms-25-09234-f004:**
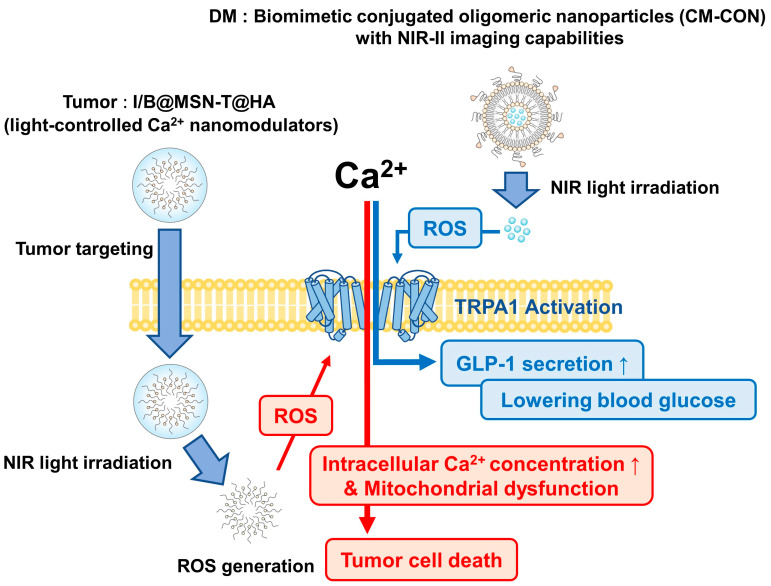
Application of nanotechnology on tumor and diabetes treatment targeting TRPA1 activation. Tumor: NIR light-irradiated silica nanoparticles (MSNs) with tumor cell targeting properties, I/B@MSN-T@HA (light-controlled Ca^2+^ na-modulator), trigger ROS generation and Ca^2+^ influx through TRPA1 channels. This approach maximizes intracellular Ca^2+^ concentration, resulting in mitochondrial dysfunction, ATP reduction, and tumor cell death (red arrows); ROS generation, and photothermal effects are exposed to an 808 nm laser, they generate ROS and local heat to activate the TRPA1 ion channel. Ca^2+^ introduced into cells by activation increases GLP-1 secretion and lowers blood sugar levels (blue arrows).

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
