# Peer review of "TRPA1-Related Diseases and Applications of Nanotherapy"

_ijms, 2024, doi:10.3390/ijms25179234_

Round 1

Reviewer 1 Report

Comments and Suggestions for Authors

This review article is dedicated to TRPA1 cation channels, and describes their structure, their physiological and pathological implications. In addition, there is a valuable section on the use of different nanotechnologies suitable for the modulation of these channels in different diseases.
The article is well written and contains a summary of important contributions in the field. However, the references are not updated, with only one 2024 reference and just a few in 2023. Since this compendium could help other people working on TRPA1, I would ask the authors to work a bit more on new references. Also missing a section dedicated to main agonist and antagonist modulators that will add value to the revision.

Among others, as important references that are not cited, I might suggest:

Current Neuropharmacology 2024, 22, 6-14.

European Medicinal Chemistry 2023, 257, 115392.

Molecular Neurobiology, 2023, 60, 5578.

Journal of Cardiovascular pharmacology 2024, 84, 10-17.

Int. J. Mol. Sci. 2022, 23, 4529.

Eur. J. Pharmacol. 2023, 959, 176088.

Nano Lett. 2023, 23, 10608.

Nat. Commun. 2022, 13, 6304.

Frontiers Pharmacol. 2022, 13, 1014041.

Comments on the Quality of English Language

English language is fine

Author Response

Comments 1:

This review article is dedicated to TRPA1 cation channels, and describes their structure, their physiological and pathological implications. In addition, there is a valuable section on the use of different nanotechnologies suitable for the modulation of these channels in different diseases.
The article is well written and contains a summary of important contributions in the field. However, the references are not updated, with only one 2024 reference and just a few in 2023. Since this compendium could help other people working on TRPA1, I would ask the authors to work a bit more on new references. 

Response 1:

I would like to express my deepest gratitude for your valuable comments. Before writing this review, I was asked to update the references to 50% or more of the articles published in the last 5 years. So I tried my best to describe the latest information, and as a result, I was able to update 60% of the references in the review to articles published in the last 5 years. The 2024 and 2023 papers on TRPA1 that I targeted are still controversial and ambiguous, so I couldn't cite many of them. I would appreciate it if you could be generous in this regard.

Comments 2:

Also missing a section dedicated to main agonist and antagonist modulators that will add value to the revision.

Response 2:

I also thought about this section, but several papers have already published it in detail, including tables. So I organized the section a bit and focused on specific diseases related to TRPA1 and nanotechnology that could be an alternative. So I would appreciate it if you could consider the fact that major agonists and antagonist modulators were not mentioned.

Reviewer 2 Report

Comments and Suggestions for Authors

The enclosed manuscript provides a sophisticated review of the role and functions of TRPA1, followed by an extended discussion on nanoparticle-based therapeutic strategies targeting TRPA1. The author listed most TRPA1-related diseases and suggested the functions of TRPA1 in different situations. The focus suddenly changed to making TRPA1 a therapeutic target for respective diseases, particularly using nanoparticles. In general, the contents are well-organized, and the topic is well-comprehensive. The existing format is nearly acceptable for publication, but some minor issues can further polish the submitted work. 

1) The rationale of the review is ambiguous. The introduction did not highlight the necessity of focusing on TRPA1; instead, it provided a very general introduction to all TRP family members. It is unclear why TRPA (or specifically TRPA1) has been selected for deep discussion. There are quite a number of publications that can be found related to TRP proteins, so it is surprising to see that the author says the significance remains unclear. 

2) It would be more integrated if the diseases mentioned in the ch.4 correlate to the suggested treatments in ch.6. Otherwise, the reading flow of the review article seems to be a bit awkward to suddenly switch from the molecule to the drug delivery system. It is still unclear if the author intends to make use of the transportation function of TRPA1 to facilitate or enhance the nanoparticle internalization or affinity. A better explanation of the logic of applying such a molecular target is expected. 

3) It is important to clearly state whether the expression of TRPA1 is favored in pathogenic events versus using it to facilitate nanoparticle targeting. The current setting is ambiguous to tell what the authors expect TRPA1 to do in nanoparticle-based treatments. 

Comments on the Quality of English Language

A moderate English editing service might be needed to make the reading experience better for readers. 

Author Response

Comments 1:

1) The rationale of the review is ambiguous. The introduction did not highlight the necessity of focusing on TRPA1; instead, it provided a very general introduction to all TRP family members. It is unclear why TRPA (or specifically TRPA1) has been selected for deep discussion. There are quite a number of publications that can be found related to TRP proteins, so it is surprising to see that the author says the significance remains unclear. 

Response 1:

I sincerely appreciate your detailed comments. I would like to address the general description of TRP channels in the first part of the introduction. I thought that a general knowledge of the supergroup called TRPs was necessary to help a more systematic understanding of TRPA1, a subgroup of TRPs. However, compared to the general description of TRP channels, the amount of information on the importance and necessity of TRPA1 in the latter part of the introduction was relatively small. Therefore, I think you may have thought that it was unclear why TRPA1 was chosen. I apologize for the lack of order in my description.

Comments 2:

2) It would be more integrated if the diseases mentioned in the ch.4 correlate to the suggested treatments in ch.6. Otherwise, the reading flow of the review article seems to be a bit awkward to suddenly switch from the molecule to the drug delivery system. It is still unclear if the author intends to make use of the transportation function of TRPA1 to facilitate or enhance the nanoparticle internalization or affinity. A better explanation of the logic of applying such a molecular target is expected. 

Response 2:

Thank you for your very thoughtful and good comments. As you mentioned, I initially focused on the correlation between the diseases mentioned in Chapter 4 and the proposed treatments in Chapter 6. However, compared to the range of diseases associated with TRPA1 that are currently known, nanotechnology-based treatments are still in their infancy, and it is difficult to explain the clear mechanisms. So, first, I wanted to introduce many diseases associated with TRPA1 to inform people that TRPA1 is valuable as a therapeutic target. Second, I wanted to suggest that nanotechnology-based treatments may be more effective than existing treatments. Therefore, I included a figure schematically illustrating the mechanisms for all treatments using nanotechnology. Although the authors of the references have expressed their claims in a simple and understandable manner, there are still many ambiguous parts, so I hope you understand that I cannot explain it as clearly as you have suggested.

Comments 3:

3) It is important to clearly state whether the expression of TRPA1 is favored in pathogenic events versus using it to facilitate nanoparticle targeting. The current setting is ambiguous to tell what the authors expect TRPA1 to do in nanoparticle-based treatments. 

Response 3:

I appreciate your critical and valuable comments, and they are very helpful to me. As you can see in the references, the effectiveness of nanotechnology-based therapies compared to conventional therapies has been proven. However, the detailed mechanisms, such as whether they act directly on TRPA1 or on signaling pathways involving TRPA1, are still unclear. In conclusion, I think that by describing the effectiveness of nanotechnology-based drug and molecular delivery systems, we can develop ideas for more advanced therapies. In addition, as more research results on the mechanism of action are derived, the role of TRPA1 in nanoparticle-based therapies will be elucidated, as you mentioned. As this is still in its initial stages, I hope for broad understanding in this regard.

Comments 4:

A moderate English editing service might be needed to make the reading experience better for readers. 

Response 4:

Before submitting this manuscript, I had it proofread by a native English-speaking physiology expert at a proofreading firm recommended by my institution.